# Effects of Differences of Breakfast Styles, Such as Japanese and Western Breakfasts, on Eating Habits

**DOI:** 10.3390/nu14235143

**Published:** 2022-12-02

**Authors:** Mai Kuwahara, Yu Tahara, Takahiko Suiko, Yuki Nagamori, Shigenobu Shibata

**Affiliations:** 1Laboratory of Physiology and Pharmacology, School of Advanced Science and Engineering, Waseda University, Shinjuku-ku, Tokyo 162-8480, Japan; 2Graduate School of Biomedical and Health Sciences, Hiroshima University, Hiroshima 734-0037, Japan; 3Research and Development Headquarters, Lion Corporation, Edogawa, Tokyo 132-0035, Japan

**Keywords:** breakfast, Japanese breakfast, western breakfast, cereal breakfast, eating habits, protein, balanced diet

## Abstract

A balanced diet and protein source intake are reportedly good for health. However, many people skip breakfast or have a light breakfast. Thus, this study aimed to examine the influence of breakfast styles on eating habits among Japanese workers, including traditional Japanese-style breakfast (JB), a pattern in which Japanese foods are eaten; Japanese-Western-style breakfast (J–W B), a pattern in which Japanese and Western foods are eaten alternately; Western-style breakfast (WB), a pattern in which Western foods are eaten; and cereal-style breakfast (CB), a pattern in which cereal is eaten. We hypothesized that breakfast style may be related to good eating habits. Data from 4274 respondents (67.97% male, 33.03% female, age 48.12 ± 0.19 years), excluding night shift workers and breakfast absentees out of a total of 5535 respondents, were analyzed. The results suggest that Japanese food is linked to the intake of protein sources such as fish, eggs, and soy. Furthermore, it was suggested that Japanese food breakfast is effective for good eating habits, such as not eating irregular amounts of food, not eating snacks, not drinking sweet juices, and having a balanced diet.

## 1. Introduction

Healthy eating habits are an important factor for good health [1,2,3]. The Japanese diet, known as washoku, is attracting worldwide attention as a healthy diet. In 2013, traditional Japanese food was included in the UNESCO Intangible Cultural Heritage list. Consumption of Japanese food is characterized by (1) frequent consumption of a wide variety of seasonal foods, including vegetables, fish, soy products, seaweed, fruits, mushrooms, and potatoes; (2) food preparation based on large amounts of high-quality water; and (3) balanced nutrition [4,5]. Japanese foods are known to have a positive impact on health as the Japanese diet is associated with a lower risk of non-alcoholic fatty liver disease in Japanese men [6] and a lower incidence of sarcopenia in both men and women [7]. In addition, the traditional Japanese diet has been introduced as a healthy diet for the prevention of cardiovascular disease (CVD) in Western countries, and a modified Japanese diet as for the DASH diet, reduced hypertension [8,9]. By contrast, the results of dietary patterns of Japanese adults in the 2012 National Health and Nutrition Survey showed that the bread pattern was associated with higher total cholesterol levels in women and higher low-density lipoprotein cholesterol levels in both men and women [10]. Soy-based foods such as fermented miso and tofu, which are typical of the Japanese diet, are known to reduce blood pressure and blood glucose levels [11,12]. The Japanese diet has long attracted interest from other countries, mainly because of its potential contribution to the low rates of coronary artery disease and increased life expectancy [13].

Breakfast intake is associated with lower BMI [14,15] and lower prevalence of chronic diseases such as coronary heart disease [16] and type 2 diabetes [17,18], with a positive impact on longevity [19].

The Japanese style of breakfast intake includes rice and miso soup as the main ingredients; the Western style of breakfast intake includes bread, milk/soy, coffee/tea, and Western/Chinese soup; and the Japanese/Western style involves eating Japanese and Western food alternately depending on the day. Meanwhile, the cereal-breakfast style is considered convenient because people commonly tend to be busy or not hungry in the morning [20].

In Japan, people who eat breakfast often have a healthy lifestyle, including a healthy diet and regular exercise habits. A higher ratio of total energy intake to fat energy intake in the morning may result in lower absolute energy intake throughout the day [21], and breakfast intake has been demonstrated to prevent excessive energy intake. Additionally, increased protein source intake at breakfast during resistance exercise training is positively associated with muscle hypertrophy in middle-aged women [22]. However, the actual problem is that many working Japanese people tend to miss breakfast or finish with simple meals. Skipping staple foods at breakfast and excessive intake of staple foods at lunch and dinner may be associated with poor sleep–wake regularity [23]. Skipping breakfast was also associated with obesity, suggesting that skipping breakfast was a predictor of being overweight among men aged 20–49 years [24]. Evening-types, who are more likely to skip breakfast, have also been demonstrated to be associated with a higher risk of obesity and cardiometabolic disease and poorer health than morning-types. Furthermore, evening-types had a higher tendency to consume harmful foods such as sugary drinks, alcohol, and chocolate than healthy foods such as fish, fruit, and vegetables. Hence, chronotype was independently associated with abdominal obesity and visceral fat [25,26,27,28]. Breakfast is the most important meal of the day for children and adults to maintain their morning chronotype.

Consuming breakfast is important; however, dietary differences, such as eating out, high salt intake, and insufficient chewing, are also important factors for health. For example, eating out more frequently has been reported to affect the quality of dietary intake [29], particularly in relation to inadequate intake of dietary fiber, vitamin C, minerals (iron, magnesium, and potassium), low vegetable intake, and high oil and fat intake [30]. High salt intake in the diet increases blood pressure and the risk of kidney disease [31,32]. Salt restriction has been demonstrated to stabilize blood pressure [33]. As Japan is a hyper-aged society compared to other countries, preventing hypertension among the Japanese population is increasingly important as older age is associated with a higher risk of CVD, which reduces healthy life expectancy [9].

Although it is widely known that Japanese food has a positive impact on health and that breakfast is important, the impact of different breakfast styles on health remains unknown. This study aimed to investigate the impact of breakfast styles on eating habits among working Japanese individuals, including Japanese-style breakfast (JB), a pattern in which Japanese food items such as rice are included; Japanese–Western-style breakfast (J–WB), a pattern in which Japanese and Western food items are eaten alternately; Western-style breakfast (WB), a pattern in which bread is included; and cereal breakfast (CB), a pattern in which cereal is included for breakfast. We hypothesized that good eating habits are related to breakfast style.

## 2. Materials and Methods

### 2.1. Ethical Approval

This study was approved by the Ethics Review Committee on Research with Human Subjects of Waseda University and the Lion Corporation (No. 2020-046 and No. 352, respectively) and followed the guidelines laid down in the Declaration of Helsinki.

### 2.2. Target Population and Data Collection

The target population comprised Japanese male and female workers aged 20–69 years living in Japan (from Hokkaido to Okinawa). The survey respondents were recruited from the Lion Corporation, Tokyo, Japan. The Internet-based questionnaire survey was conducted from 18 to 22 June 2021.

### 2.3. Questionnaire

The online questionnaire consisted of 18–21 items regarding basic information on the respondent, such as sex, age, body mass index (BMI), residential area, and job description, and 60–82 questions on the respondent’s lifestyle, including work and personal life stress, health and personality, diet, and COVID-19 status. The study was designed by the Lion Corporation and the Shibata Laboratory at Waseda University and conducted by a WEB search company ordered by Lion Corporation, and the purpose of the questionnaire was to determine the lifestyle, eating habits, exercise habits, and health awareness of the Japanese workers.

The questionnaire was designed to take approximately 30 min to answer a total of 103 questions, but some people took more time to finish. Moreover, because this was an Internet-based questionnaire, people who met any of the following criteria were excluded from the survey: age ≤19 years and ≥70 years; unemployed; officers (directors and executive directors); working less than 2 working days in a week; and did not respond to any of the questions. Residential areas were equally included from northern to southern areas and from rural to urban areas. Jobs were equally included from civil servants to private companies and large to small companies. The number of participants who completely answered all questionnaires was 5534 (male *n*= 3849, female *n* = 1685) people. The results of the questionnaire responses from male and female workers who worked night shifts (22:00–5:00 the following day) (*n* = 808, male *n* = 614, female *n* = 194) were excluded, and residual participants were (*n* = 4726, male *n* = 3235, female *n* = 1491). To examine the impact of breakfast consumption on eating habits, data from those who reported missing breakfast (*n* = 452; male *n* = 330, female *n* = 122) were excluded. Thus, the ratio of breakfast missing was 10% for males and 8% for females, and these data had slightly lower values than the results (male 14%, female 9%) of Japanese adults in the 2018 National Health and Nutrition Survey. The final numbers for analysis in this report were *n* = 4274 (male *n* = 2905, female *n* = 1369) (Appendix A). The data from these questionnaires were on sex, age, BMI, and breakfast style (JB, J-W B, WB, and CB). The intake of food items in the usual diet (7 levels) and eating habits in the usual diet (7 levels) was assessed using questionnaires (Appendix A).

### 2.4. Statistical Analysis

Statistical analyses were conducted using IBM SPSS Statistics (version 28; IBM Japan, Ltd., Tokyo, Japan). A non-parametric analysis (Mann–Whitney U test) was used to compare the results of breakfast style, respondent characteristics, and eating habits between men and women. A multiple regression analysis was conducted to examine the impact of breakfast style on eating habits. Between-group comparisons of breakfast style were made using the Kruskal–Wallis test and Dunn’s test. Statistical significance was set at *p* < 0.05. Data are expressed as mean and standard error.

## 3. Results

### 3.1. Participant Characteristics

The participants (*n* = 4274) comprised 67.97% of males and 33.03% of females (Appendix A). The mean age and BMI of the participants were 48.12 ± 0.19 years old and 22.68 ± 0.05 kg/m^2^, respectively. Daily intake of protein sources other than fish was significantly higher in women than in men. The daily intake of vegetables, fruits, and snacks was significantly higher in women than in men, while the intake of juice and alcohol was lower in women than in men. Regarding eating habits, smaller scores indicate healthier eating habits. Women were shown to have healthier eating habits than men in the categories ‘eat less food’, ‘eat out less’, and ‘don’t eat too much salt’.

### 3.2. Relationship between Breakfast Style and Usual Protein Source Intake

To examine the association between breakfast style and daily protein source intake, a multiple regression analysis was performed (Table 1). Protein source intake scores were entered as the objective variable, breakfast style (JB, J-W B, WB, or CB) as the explanatory variable, and sex, age, and BMI as adjustment factors. Similarly, the multiple regression analysis revealed that JB was associated with fish, egg, and bean intake in the usual diet. By contrast, WB and CB were related to the intake of dairy products in the usual diet. The mean scores (on a scale of 5) of the participants’ protein source intake were averaged and compared between the groups. Kruskal–Wallis and Dunn’s tests were used for between-group comparisons (Figure 1). The intake frequency of fish and eggs was high in the JB and J-W B groups. The intake frequency of beans was high in the JB group. The intake frequency of dairy products was higher in the WB and CB groups than in the JB and J-W B groups.

### 3.3. Relationship between Breakfast Style and Usual Food Intake

A multiple regression analysis was conducted to examine the relationship between breakfast style and staple foods, vegetables, fruits, and beverages (Table 2). Food and beverage intake scores were entered as objective variables, breakfast style (JB, J-W B, WB, or CB) as explanatory variables, and sex, age, and BMI as adjustment factors. Multiple regression analysis revealed that consuming a JB or WB was related to consuming staple food in their usual diet. Consuming JB is associated with the consumption of vegetables and fruits in the usual diet. The consumption of WB is associated with the consumption of fruits and snacks in the usual diet. Participants’ scores for whether they consumed staple foods (on a scale of 5) and their average score for food intake (on a scale of 5) were averaged and compared between the groups. The Kruskal–Wallis test and Dunn’s tests were used for this between-group comparison (Figure 2). The staple food intake frequency was higher in the JB and WB groups and lower in the J-W B and CB groups. The consumption of snacks and juice frequency was lower in the JB, J-W B, and WB groups, in that order. Meanwhile, the frequency of alcohol consumption was higher in the JB and J-W B groups.

### 3.4. Relationship between Breakfast Style and Usual Eating Habits

A multiple regression analysis was performed to examine the relationship between breakfast style and eating habits (Table 3). Eating habit scores were entered as the objective variable, breakfast style (JB, J-W B, WB, or CB) as the explanatory variable, and sex, age, and BMI as adjustment factors. Multiple regression analysis revealed that consuming a Japanese breakfast had an impact on consuming regular amounts of food, not eating too much food in one sitting, having balanced nutrition, and not having excessive salt intake. Participants’ eating habit scores (on a scale of 7) were averaged and compared between the groups. Kruskal–Wallis and Dunn’s tests were used for between-group comparisons (Figure 3). Consuming irregular amounts of food was lower in the JB group than in the J-W B group. The frequency of unbalanced nutrition and eating full meals was lower in the JB group than in the WB group.

## 4. Discussion

To the best of our knowledge, this is the first cross-sectional study to assess the association between breakfast type and eating habits among Japanese workers. Our results demonstrate that even a light breakfast, such as cereals, has a positive impact on eating habits, but that JB, in particular, has a positive impact on eating habits throughout the day, not just in the morning.

### 4.1. Risks of Skipping Breakfast

Many previous studies have examined breakfast skipping. Skipping breakfast has been demonstrated to affect elements of energy balance [34], although it has also been suggested to be a predictor of being overweight [34,35]. Weight control is often cited as the reason for not eating breakfast. Weight control has been reported to be associated with unhealthy behaviors, poor diet, and lower physical activity [36,37,38,39], and is also associated with higher metabolic risk, that is, higher BMI. Here, not skipping breakfast was linked to vegetable intake and eating a balanced diet. We focused on the type of breakfast rather than skipping breakfast because skipping breakfast is considered unhealthy.

### 4.2. Japanese Food Styles and Their Effects

The study primarily aimed to examine what kind of breakfast, if any, would have health benefits. Previous studies have reported that soy, traditionally consumed in Japan, is expected to prevent lifestyle-related diseases such as stroke and coronary heart disease [40]. Furthermore, eating fish regularly has also been reported to prevent lifestyle-related diseases, such as hypertension and diabetes [41,42]. Japanese food is generally rich in soy, seafood, miso, soy sauce, vegetables, potatoes, and mushrooms; therefore, consuming a Japanese breakfast is a good way to consume these foods. The consumption of seafood provides dietary amino acids, which play an important role in protein synthesis [43]. Additionally, soy provides an important source of protein, dietary fiber, vitamins such as folate, and minerals such as iron, zinc, and magnesium. Increasing the intake of these nutrients can enhance the quality of the diet. Saponins and tannins in pulses also have antioxidant and anti-carcinogenic properties, indicating that pulses may have significant anti-cancer properties [44]. The Japan National Health and Nutrition Survey 2012 revealed that more than 95% of the participants met recommended levels of protein suggested by the Dietary Reference Intake in Japan [45]. Thus, the current study suggests that the Japanese breakfast leads to not only an increased volume of protein source intake but also to a variety of protein sources such as fish, eggs, and soy throughout the day.

The main sources of energy in the diet should be complex carbohydrates, and it is not a bad sign if the reduction in protein source intake is compensated by carbohydrates with a small addition of fat. On the other hand, if the proportion of protein sources in the diet decreases, too much excess energy intake from non-protein energy nutrients (carbohydrates and fats) may be driven to compensate for the energy deficit caused by reduced protein source intake [46,47]. Interestingly, Japan National Health and Nutrition Surveys during the period from 1995 to 2016 have demonstrated that total energy intake showed a decreasing trend in both males and females, but energy intake from protein decreased and energy intake (%) from fat increased in both sexes [48]. Thus, excessive intake of carbohydrates and fats leads to obesity. Among proteins, soy-based foods, such as miso and tofu, are known to reduce blood pressure and blood glucose levels [11,12], and fish oil, DHA, and EPA found in fish can reduce or improve the risk of CVD [49,50,51,52]. Sugar intake causes CVD, type 2 diabetes, obesity, metabolic syndrome, and inflammation [53,54]. Furthermore, the consumption of Japanese food is negatively related to the intake of sugar-sweetened drinks such as juices. Consuming Japanese food may prevent excessive sugar intake; thus, it may have a positive impact on health. In this study, the consumption of Japanese food was positively related to alcoholic beverage consumption. However, the mechanism behind the association between the consumption of Japanese food and alcohol intake is unknown. The current study suggests that traditional Japanese diets lead to a high intake of fish and beans. Consuming a Japanese diet has also been epidemiologically associated with a lower risk of CVD [55,56] and beneficial for blood pressure if salt intake is optimized [10]. Here, no association between excessive salt intake and consumption of Japanese breakfast food was identified.

### 4.3. Eating Habits and Breakfast Intake Style

Eating habits and breakfast intake styles are related, and these have a direct connection to health. When we consume food with irregular rhythms, the thermic effect of food is significantly less than with normal dietary patterns, and the increased responsiveness to glucose may have adverse effects on health, such as causing obesity and reduced metabolism [57,58], Metabolic syndrome has been linked to BMI, and cardiovascular metabolic risk factors such as blood pressure have also been suggested to be at higher risk [59]. A healthy balanced diet provides the human body with energy (macronutrients) and all necessary micronutrients (vitamins, minerals, protein-producing amino acids, and omega fatty acids) required for the maintenance of all metabolic processes [60]. This suggests the importance of a healthy and balanced diet. Several studies have suggested that a diet rich in antioxidants and anti-inflammatory components, such as fruits, nuts, vegetables, and fish, can reduce the risk of age-related cognitive decline and various neurodegenerative diseases [61]. A balanced and nutritious diet is important for maintaining good health, and the current study suggests that a balanced diet can be achieved by JB. Among American adults, the number of daily snacks increased by approximately one snack per day between 1977 and 2006 [62]. Thus, owing to the prevalence of snacks in today’s society, energy-dense and low-quality snacks, including high-fat and high-sugar snack foods, are associated with an increased risk of obesity and CVD [62,63,64]. High-fiber snacks may have a positive effect, such as lowering blood glucose levels [65], and high-protein snacks may promote further satiety [64] and reduce overconsumption at the next meal [64]; therefore, what you eat may or may not be detrimental to your health. Being full is one of the important psychobiological mechanisms that function to inhibit post-intake of food and drinks and is an important factor in reducing overconsumption that can lead to being overweight and obesity [64,66], in addition to the fact that snacking still leads to obesity. Here, a negative correlation between consuming a JB and consuming snacks was identified, suggesting that consuming Japanese food for breakfast tends to make people less likely to consume snacks. On the other hand, snacking plays an important role in replenishing what is lacking in nutrition from regular meals. In the current study, participants were workers, and it is possible to eat snacks for replenishing lacking energy between meal times.

### 4.4. Breakfast Realities and History

Many Japanese people consider it ideal to consume Japanese food such as rice and miso soup for breakfast. In reality, most people consume Western (bread-based) breakfasts. One of the reasons behind the gap between the ideal and reality is that many people believe that they do not have time in the morning [67]. This survey was conducted on only 107 Japanese adults in Tokyo. On the other hand, the present results revealed that JB and WB were 28% and 41%. In addition, recently our paper demonstrated that 2571 participants were categorized by JB (31% for male, 24% for female) and WB (28% for male, 31% for female) [68]. Thus, there was not a big difference in the ratio of Japanese style and Western style breakfast. It is much easier to bake bread and spread jam than to cook rice and make side dishes and miso soup, and even easier to make a CB-style breakfast, which is completed simply by adding milk to cereal. Thus, those who can afford to consume Japanese food in the morning may be able to pay attention to a balanced diet, the amount of food they eat, and their vegetable and protein source intake.

In Japan, Western-style eating did not begin until after World War II, when Japan imported food from other countries to solve malnutrition. However, Western-style diets are said to lead to overeating and obesity, which may affect lifestyle-related diseases and other health problems [69].

### 4.5. Limitation of Current Study

There are some limitations to our study. First, food items and eating habits were collected through self-reports, and this self-reporting may have resulted in self-efficacy. Second, although there are 21 characteristics of respondence, we used only three variables in this paper (gender, age, BMI). Therefore, more variables should be used to analyze the associations. Third, “eating out” is an inadequate question, because we do not define whether eating out means nutrition balanced healthy meals or oily unhealthy fast food. Fourth, “snack intake” is also an inadequate question, because we do not define whether snack intake is for the replenishing of what is lacking or for just fun. Fifth, we want to know the association between food items/eating habits and not only breakfast style, but lunch and dinner meal style to understand the role of traditional Japanese food in healthy eating behavior.

## 5. Conclusions

Different types of breakfast have different effects on eating habits, such as daily intake of items and eating habits. The intake of JB may have a positive impact on eating habits, as it is related to the regularity and quantity of fish, eggs, and beans in the usual diet and balanced diet. Meanwhile, the intake of WB and CB is positively related to the intake of dairy products in the usual diet, and consuming WB is negatively related to the intake of soy, so a JB and occasionally a WB, may lead to a more balanced diet.

## Figures and Tables

**Figure 1 nutrients-14-05143-f001:**
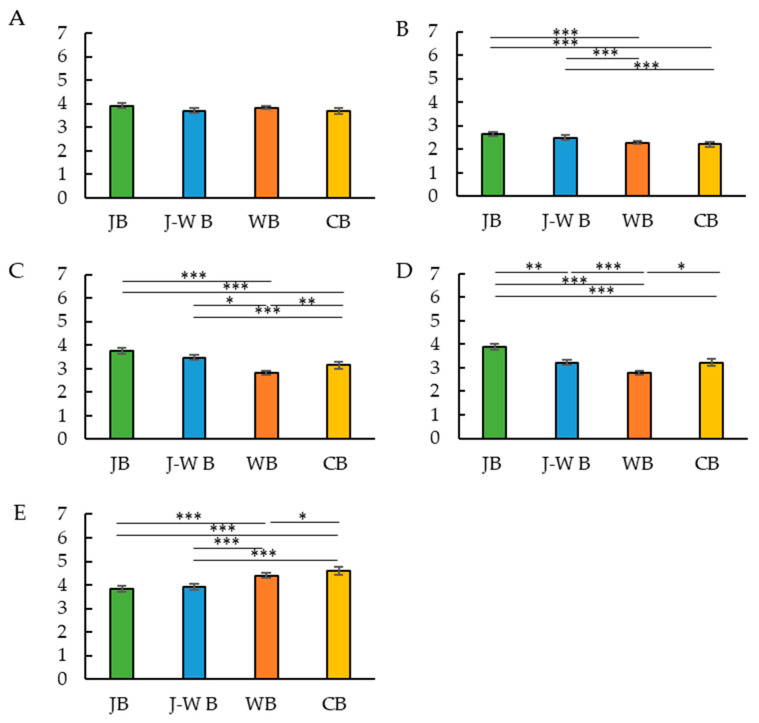
Breakfast style and usual protein source intake score. Breakfast style and meat score (**A**). Breakfast style and fish score (**B**). Breakfast style and eggs score (**C**). Breakfast style and beans score (**D**). Breakfast style and dairy products score (**E**). * *p* < 0.05, ** *p* < 0.005, *** *p* < 0.001 (Dunn’s test).

**Figure 2 nutrients-14-05143-f002:**
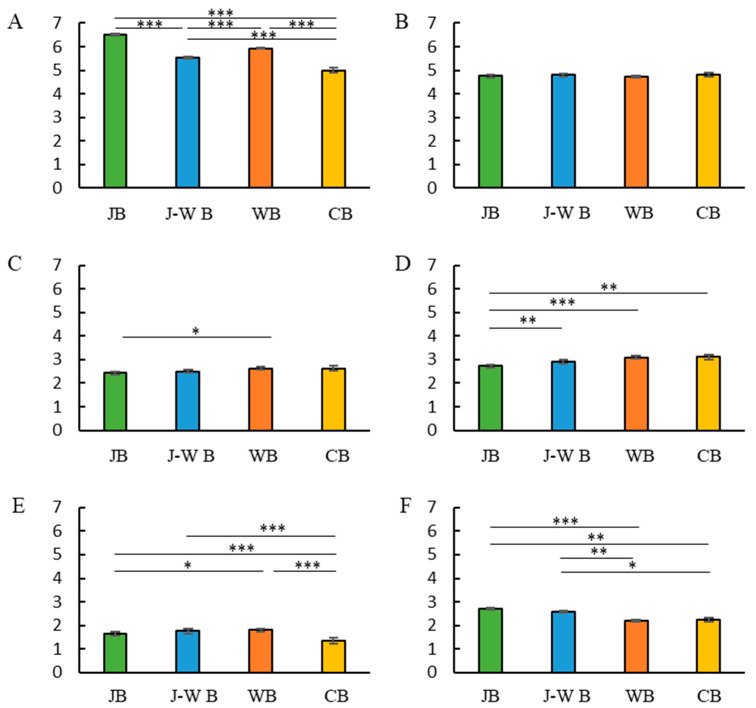
Breakfast style and usual food score. Breakfast style and staple food score (**A**). Breakfast style and vegetable score (**B**). Breakfast style and fruits score (**C**). Breakfast style and snack score (**D**). Breakfast style and juice score (**E**). Breakfast style and alcohol score (**F**). * *p* < 0.05, ** *p* < 0.005, *** *p* < 0.001 (Dunn’s test).

**Figure 3 nutrients-14-05143-f003:**
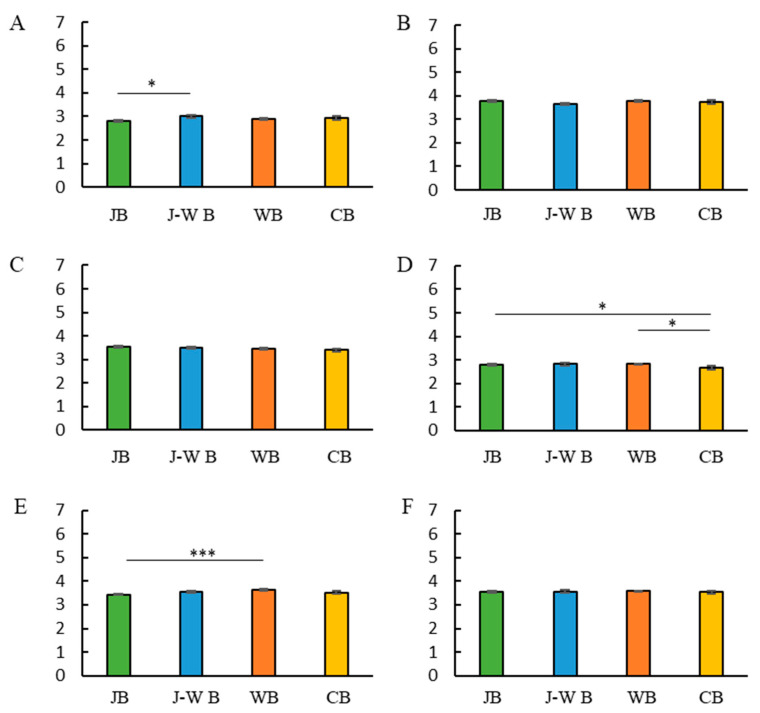
Breakfast style and usual eating habits. Breakfast style and irregular amount of food score (**A**). Breakfast style and less chewing (**B**). Breakfast style and large amounts of food (**C**). Breakfast style and frequent eating out (**D**). Breakfast style and nutritional imbalance (**E**). Breakfast style and excessive salt intake (**F**). * *p* < 0.05, *** *p* < 0.001 (Dunn’s test).

**Table 1 nutrients-14-05143-t001:** Results of multiple regression analysis of the relationship between breakfast style and protein source intake.

	JB		J-W B		WB		CB		R^2^	F
	β	*p*	β	*p*	β	*p*	β	*p*		
Meat	0.087	*p* < 0.005	0.050	*p* < 0.05	0.069	*p* < 0.05	0.038	0.053	0.017	11.516
Fish	0.163	*p* < 0.005	0.124	*p* < 0.005	0.075	*p* < 0.005	0.039	*p* < 0.05	0.016	10.702
Eggs	0.193	*p* < 0.005	0.138	*p* < 0.005	0.132	*p* < 0.001	0.041	*p* < 0.05	0.021	14.348
Beans	0.188	*p* < 0.005	0.104	*p* < 0.005	0.023	0.368	0.067	*p* < 0.001	0.036	25.139
Dairy products	0.175	*p* < 0.005	0.163	*p* < 0.005	0.274	*p* < 0.001	0.208	*p* < 0.001	0.082	60.236

R^2^ indicates the contribution ratio and β the standardized partial regression coefficient.

**Table 2 nutrients-14-05143-t002:** Results of the multiple regression analysis of the relationship between breakfast style and food intake.

	JB		J-W B		WB		CB		R^2^	F
	β	*p*	β	*p*	β	*p*	β	*p*		
Staple food	0.246	*p* < 0.001	0.139	*p* < 0.001	0.285	*p* < 0.001	0.009	0.640	0.052	37.208
Vegetables	0.181	*p* < 0.001	0.164	*p* < 0.001	0.170	*p* < 0.001	0.123	*p* < 0.001	0.068	48.914
Fruits	0.181	*p* < 0.001	0.173	*p* < 0.001	0.226	*p* < 0.001	0.147	*p* < 0.001	0.049	34.466
Snack	0.064	*p* < 0.05	0.079	*p* < 0.001	0.134	*p* < 0.001	0.072	*p* < 0.001	0.073	53.147
Juice	−0.039	0.105	−0.008	0.726	0.008	0.765	−0.068	*p* < 0.001	0.014	9.230
Drinking	0.005	0.843	−0.004	0.864	−0.089	*p* < 0.001	−0.020	0.293	0.089	65.962

R^2^ indicates the contribution ratio and β the standardized partial regression coefficient.

**Table 3 nutrients-14-05143-t003:** Results of multiple regression analysis of the relationship between breakfast style and eating habits.

	JB		J-W B		WB		CB		R^2^	F
	β	*p*	β	*p*	β	*p*	β	*p*		
Consumption of irregular amounts of food	−0.265	*p* < 0.001	−0.194	*p* < 0.001	−0.262	*p* < 0.001	−0.164	*p* < 0.001	0.057	40.450
Less chewing	−0.068	*p* < 0.005	−0.099	*p* < 0.001	−0.073	*p* < 0.005	−0.051	*p* < 0.05	0.019	13.184
Consumption of large amounts of food	−0.013	*p* < 0.001	−0.115	*p* < 0.001	−0.130	*p* < 0.001	−0.089	*p* < 0.001	0.058	41.115
Frequent eating out	−0.087	*p* < 0.001	−0.070	*p* < 0.001	−0.074	*p* < 0.005	−0.080	*p* < 0.001	0.041	28.596
Nutritional imbalance	−0.202	*p* < 0.001	−0.162	*p* < 0.001	−0.157	*p* < 0.001	−0.125	*p* < 0.001	0.029	20.289
Excessive salt intake	−0.120	*p* < 0.001	−0.102	*p* < 0.001	−0.112	*p* < 0.001	−0.075	*p* < 0.001	0.019	12.697

R^2^ indicates the contribution ratio and β the standardized partial regression coefficient.

## Data Availability

Data will be sent upon request from the corresponding author. The data are not publicly available because of patent preparation.

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
