# Peer review of "Effects of Differences of Breakfast Styles, Such as Japanese and Western Breakfasts, on Eating Habits"

_nutrients, 2022, doi:10.3390/nu14235143_

Round 1

Reviewer 1 Report

Dear Aurhors, Dear Editors,

I found it interesting to read the manuscript on Japanese breakfasts and eating habits, but the reading leaves me somewhat unsatisfied. The main point is that there are some shortcomings regarding knowledge of the science of human nutrition. The introduction and discussion contain various inaccuracies, unauthorized generalizations and a lack of precision in the formulation of thoughts. Even with regard to the title and keywords, I have a concern as to whether the authors see the difference between "eating habits" and "dietary habits"? In my opinion, one term is enough. The second issue is, as it were, the "glorified" consumption of protein, which in the populations of highly developed countries is too excessive in relation to needs. Sure, there are differences in the amount of consumption between countries, but FAO data shows that in Germany, for example, the aggregate consumption of meat and fish (with meat predominating) is about 90 kg, and in Japan (with fish predominating) - about 100 kg. The fact is that the traditional Japanese diet is health-promoting (soy, legumes, fish, algae, vegetables), but the authors themselves write in subsection 4.4 that most Japanese have WB breakfasts.

Introduction - specific comments:

L.28. "Japanese food" - I suggest adding "traditional".

L.29. Should be "Japanese food" + consumption.

L.30. There are vegetables twice in item (1).

L.35-37. You can simply add that it is about the DASH diet.

L.38-39. it is necessary to specify which population the study was about [10].

L.46-47. It is necessary to cite the source for such a radical statement.

L.49-50. the study is about the adult population, there is no need to cite the data for children and adolescents, especially since source 17 is from 1986. besides, this whole paragraph is quite superficial, since light WB breakfasts are associated with the fact that lunchtime starts at 12 o'clock in many Western countries.

L.59-73. this paragraph is about Japan, and already in L.59 this needs to be added. L.64. It is worth adding "in women" before citing the source.

Methods. In my opinion, it is necessary to explain the methodology of the study simply, without any ambiguity and in chronological order, including in particular:

- a statement that the study was designed and carried out by the Lion Corporation (that's my guess) and its purpose was ... (based on L.107-110);

- details of the sampling (in addition to the participation and exclusion criteria) and whether the sample by any/all characteristics was nationally representative;

- To collect in one place the inclusion and exclusion criteria for the study, as they are contained in different paragraphs (L.111-114, L.120-122, L.127-128)

- How should we understand the sentence in L.115. (we were able to collect the final results from... )? Does it mean that more people participated in the entire survey of the Lion Corporation? What proportion of the total sample accounted for 4726 questionnaires (L.118);

- Among this population, 452 people do not eat breakfast and were excluded from the survey - therefore, the reader is interested in the characteristics of the reduced number of n=4274 (L.128) and it is worth including in the supplementary materials. If the analysis of the results, however, included n=4726, then there is a serious methodological error.

And a few more minor comments:

L.101. The survey respondents were recruited from (?) the Lion Corporation; or by (?) the LC.

L.105. "questionnaire consisted of 18-21 items regarding the basic information on the respondent" - does this mean that max. 21 characteristics of the respondent were specified? - this is a very large number, especially compared to only 3 variables used in the manuscript (gender, age, BMI) 

L.107. "and 60-82 questions on the respondent's lifestyle...". - do I understand correctly that the entire questionnaire at most could contain 103 questions and, according to the authors, it took only 30 min to complete it?  

Supplemental Table 4. Instead of "food items" it should be "eating habits".

3. Results

L.139. Among the indicated eating habits, 2 indicate quite the opposite - these are unhealthy habits. On the other hand, frequent eating out cannot be clearly evaluated, as there are many categories of food establishments in the food service market - from premium establishments to QSRs.

L.155-156 You need to decide where to write the sentence explaining the contents of Table 1 - whether in L.144-145 or under Table 1 (there is no need to repeat the same thing). The same remark applies to double explanations of Tables 2 and 3.

4. Discussion

Reading this chapter, as well as the introduction, leaves many doubts, is vague and shows that knowledge of human nutrition is not the best side of the authors. My comments:

L.219. The statement "that breakfast eaters are leaner" (and nothing else) suggests that this applies to the entire population closer to unidentified. Meanwhile, the cited source refers to a completely different population group (British schoolchildren only); besides, it was published quite a long time ago and eating habits may have changed, especially since there is a dynamic increase in the percentage of overweight and obese children.

L.235. Folate is a vitamin; it should be mentioned before minerals.

L.237. Twice the word "beans" should be replaced by the more precise "pulses."

L.239. Awkward sentence; fish (and others in this sentence) are not protein, they are a source of protein.

L.240-242 I cannot agree with this mental shortcut. The main source of energy in the diet is/should be complex carbohydrates, and it is not a bad sign if the reduction in protein intake is compensated by carbohydrates (with a small addition of fat).

L.244. There is no APA fatty acid, but there is EPA (eicosapentaenoic acid).

L.246. I didn't notice that sugar intake was studied - so why do the authors write "the consumption of Japanese food is negatively related to sugar intake, including juices" (snacks can be both health-promoting and not, but this was not defined in the study). Also, the custom of "Satiety feeding; Eating fruit or sweets even on a full stomach" explains nothing, since a typical questionnaire error was made, asking about two products in one question, especially since from a health perspective one is beneficial and the other is not.

L.298 The last subsection is unnecessary. First - it begins by repeating the content of L.263-265. Second - the importance of a healthy and balanced diet was not proven in the study. Third - throughout the manuscript it was proven that Japanese-style breakfasts are best for health, and here at the end the benefits of CB breakfast are indicated.

L.313. According to FAO nomenclature - "pulses" instead of "legumes."

I hope that my comments will allow the authors to improve the scientific value of the manuscript. All the best!

Reviewer 2 Report

The topic of the manuscript entitled "Effects of differences of breakfast styles, such as Japanese and Western breakfasts, on eating and dietary habits" ios interesting,and the results can give us a deep uderstanding of the JB is effective for the intake of proteins, while WB and CB are effective for the intake of dairy products. The manuscript is well organized and written, the manuscript needs some minor revision in the reference style.

Author Response

Submission ID: nutrients-2026061

Response to the Reviewers’ comments

Once again, we wish to thank the editor and reviewers for reading our manuscript so thoroughly and providing such constructive feedback. The quality of our manuscript has certainly improved as a result of these comments. Our point-by-point responses are provided below, and the necessary changes are highlighted in yellow in the revised manuscript.

Reviewer 2:

The topic of the manuscript entitled "Effects of differences of breakfast styles, such as Japanese and Western breakfasts, on eating and dietary habits" ios interesting,and the results can give us a deep uderstanding of the JB is effective for the intake of proteins, while WB and CB are effective for the intake of dairy products. The manuscript is well organized and written, the manuscript needs some minor revision in the reference style.

Response: Thank you very much for your kind comments. We carefully checked the reference style.

Round 2

Reviewer 1 Report

Dear Authors,
I have carefully followed your responses to my suggestions and read the corrections in the text (I think the word "people" is redundant in L. 120). I agree with the Authors that it is best to use the word "soy" instead of "beans" or "pulses" for the traditional Japanese diet. The revised version of the manuscript is fully satisfactory, especially with the revised methodology and human nutrition issues.

Kind regards